# A comparative analysis of important public clinical trial registries, and a proposal for an interim ideal one

**Nisha Venugopal, Gayatri Saberwal**📇*

Institute of Bioinformatics and Applied Biotechnology, Bengaluru, Karnataka, India

* gayatri@ibab.ac.in

**Data Availability Statement:** Most relevant data are within the manuscript and its Supporting Information files. Some data is available on the websites of the registries, at URLs that are

## Abstract

### Background

It is an ethical and scientific obligation to register each clinical trial, and report its results, accurately, comprehensively and on time. The WHO recognizes 17 public registries as Primary Registries, and has also introduced a set of minimal standards in the International Standards for Clinical Trial Registries (ISCTR) that primary registries need to implement. These standards are categorized into nine sections—Content, Quality and Validity, Accessibility, Unambiguous Identification, Technical Capacity, Administration and Governance, the Trial Registration Data Set (TRDS), Partner registries and Data Interchange Standards. This study compared the WHO's primary registries, and the US's ClinicalTrials.gov, to examine the implementation of ISCTR, with the aim of defining features of an interim ideal registry.

### Methods and findings

The websites of the 18 registries were evaluated for 14 features that map to one or more of the nine sections of ISCTR, and assigned scores for their variations of these features. The assessed features include the nature of the content; the number and nature of fields to conduct a search; data download formats; the nature of the audit trail; the health condition category; the documentation available on a registry website; etc. The registries received scores for their particular variation of a given feature based on a scoring rationale devised for each individual feature analysed. Overall, the registries received between 27% and 80% of the maximum score of 94. The results from our analysis were used to define a set of features of an interim ideal registry.

### Conclusions

To the best of our knowledge, this is the first study to quantify the widely divergent quality of the primary registries' compliance with the ISCTR. Even with this limited assessment, it is clear that some of the registries have much work to do, although even a few improvements would significantly improve them.

referenced in the various Supplementary Files, as relevant.

**Funding:** GS received internal institutional funds from IBAB, which is funded by the Government of Karnataka's Department of Information Technology, Biotechnology and Science & Technology (https://itbtst.karnataka.gov.in/english). There was no grant number. The funders had no role in study design, data collection and analysis, decision to publish, or preparation of the manuscript.

**Competing interests:** The authors have declared that no competing interests exist.

## Introduction

The first two calls for clinical trial registries were made in the 1970s [1]. One aimed to enhance the enrolment of patients in ongoing trials, and the other to reduce the possibility of bias in the subsequent reporting of trial results, caused by the selective publication of those with positive outcomes. Since the year 2000, trial registries have proliferated. Nevertheless, it has been a long and sustained battle by many stakeholders–activists, journals, researchers, funders, governments and the World Health Organization (WHO)–to ensure that large numbers of trials are registered [2–4]. Although the initial two aims for setting up registries continue to be among the most important uses of such databases, researchers have utilized the data in at least a dozen other ways, such as (i) analyzing characteristics such as the disease areas, medical interventions, and sponsors of Expanded Access Studies registered in the United States (US) [5]; (ii) identifying the fraction of trials that have run in the country, that had industry involvement [6]; (iii) conducting a geo-temporal analysis of the trials of novel stem cell therapies [7]; (iv) obtaining information about a trial that was not reported in the subsequent publication [8]; and (v) identifying trials being run in contravention to the law [9]. Given these numerous and diverse purposes, not initially envisaged, it is even more important that all trials are registered and reported in a timely fashion, and that all the data in each record is complete, reliable and readily accessible. In view of this, the quality of data in the databases has long been the subject of analysis and comment. These include (a) analyses of the quality of registration and missing information in trial records [10–13], (b) studies on the discrepancies in trial status for trials that are registered in more than one registry [14], and (c) reports on the phenomenon of hidden duplicates [15, 16]. Other studies have looked into the evolution of individual registries–the challenges faced, and advances made [17, 18]. Such analyses have improved our understanding of clinical trial registries globally, and reinforce the efforts for creating uniform standards for trial registration globally.

Certain high profile scandals, such as the Vioxx case in 2004 and the Paxil case in 2015 [19, 20] resulted in numerous calls to increase transparency in clinical trials and to improve the public's trust in the trials enterprise. Following the Ministerial Summit on Health Research that took place in Mexico City in November 2004, the WHO launched the International Clinical Trials Registry Platform (ICTRP) initiative in 2006 [21].

The ICTRP enables a single point of access to information regarding trials within its registry network [22], which hosts trial records from around the world. The network consists of (i) Primary Registries (PRs), (ii) Data providers, and (iii) Partner registries [23]. There are currently 17 PRs. The Data providers include the PRs and ClinicalTrials.gov (CTG), of the United States (US). All data providers need to fulfil the same criteria in terms of data collection and management. The two partner registries (i) are not required to fulfil the criteria that PRs need to; (ii) need to be affiliated with one of the PRs; and (iii) cannot directly feed data into the ICTRP [24]. Therefore, we have not included these two registries in our study. We have analyzed the 17 PRs, and CTG, and refer to them, collectively, as Primary Registries Plus, or PR+.

The WHO also developed the International Standards for Clinical Trial Registries (ISCTR) [25], which lists the minimum, and sometimes ideal, standards that PRs should adopt to ensure a basic quality of data and accessibility. These standards are in nine sections—Content; Quality and Validity; Accessibility; Unambiguous Identification; Technical Capacity; Administration and Governance; the 24-field Trial Registration Data Set (TRDS); Partner Registries; and Data Interchange Standards. Although, ICTRP stipulates that non-compliance with these standards can result in revoking of Primary Registry status, we are unaware of any case where this has happened.

It is known that users trust public registries more than those created by companies or patient groups [26]. Also, public registries are often the primary sources on which other databases are built [26]. It follows that the information in each one should be comprehensive, high

quality and available in a user-friendly fashion. Accordingly, there have been calls for (i) a comparison of such registries, to help develop suitable standards [25], and (ii) ways to improve the accessibility and content of the PR+ [27]. However, several years ago it was shown that there had been non-compliance with the WHO minimal dataset [28], and non- optimal website functionality and user experience [10, 11, 29]. Since across-the-board improvements have not taken place, this issue needs to be reiterated. However instead of undertaking a purely qualitative assessment, we drew inspiration from other researchers' scorecards. These scorecards have either been developed [30–33] or proposed [34, 35] to track whether trialists register their studies and report the results accurately, comprehensively and on time. Accordingly, we developed one to assess the PR+.

We have developed the Registries' Comparative Scorecard (the Scorecard) which rates the PR+ on certain features that map to different sections of the ISCTR (S1 Table). We then define an interim 'ideal registry' based on the best variations of each feature used by the PR+. Until such time as all the registries adopt all the standards recommended by ISCTR, the adoption of the recommended variations of each feature would be very helpful for users.

## Methods

### Data collection

We accessed the websites of the 18 PR+ between July 2019 and April 2020, inclusive. The registries were evaluated for 14 features that map to one or more of the nine sections of ISCTR mentioned above. The list of features was compiled by the authors based on literature regarding the necessity of higher quality trial registrations [36, 37], focusing on the standards listed in ISCTR [25].

All information was obtained from one or more of the following resources within each PR+ website: (i) the general pages of the site; (ii) a randomly chosen, sample interventional trial that was registered after 1 January 2019; (iii) supporting documents, if available; and (iv) where necessary and possible, via a login to do a mock registration. All analyses were performed by one author (NV) and verified by the other (GS), with differences resolved by discussion.

The sections below provide further methodological details on the data collection for, and analysis of, each of the features analysed, which have been classified based on the ISCTR section they map to. Reference URLs are available in the Supplementary files, which are referenced in the Results as relevant.

**I. Accessibility features.** We first examined the accessibility of information in the PR+. For this, we assessed several features, as follows:

**(i) The ease of obtaining the total number of trials hosted by the registry:** The method of obtaining the total number of trials hosted by each PR+ was determined. Specifically, we documented (a) whether the number was displayed on the home page, (b) if it was available after a search, or (c) whether it had to be calculated based on the number of pages of results. If there was discrepant information at different places on the site, this fact was captured.

**(ii) The existence of a Basic search function:** We examined the presence of the search function using a basic search field.

**(iii) and (iv) The number of TRDS fields, and extra fields, that can be used to conduct a search:** We documented the presence and number of (iii) TRDS fields; and (iv) Extra fields, beyond these 24 TRDS fields, that can be used to conduct a search.

**(v) The data download options:** For each PR+, we documented the file formats that are available for data download. We also captured information on whether the data on (a) one, (b) a limited number, (c) multiple, or (d) all search results can be downloaded at a time.

**II. Content or compliance with TRDS features. (i) TRDS fields and Extra fields:** Each registry provides information about a trial in two different 'views'. While conducting a search, the

user first obtains a list of trials which contains the titles, and may also contain other information. This is called the Brief view. Each trial record is available as a Brief view, and a Detailed view. The fields available in these views in each registry were documented. This information was then mapped to the 24 fields of the WHO TRDS. All additional fields were categorized as 'Extra fields'.

Among the Extra fields, we looked into the following features in further detail.

**(ii) Whether the Principal Investigator (PI) name is compulsory:** Even though the ISCTR states that the PI is the 'Contact for Scientific Queries', unless the PI delegates this task to somebody else, the PR+ have not uniformly adopted this definition, and it is not always clear if the 'Contact for Scientific Queries' reflects the PI. Therefore, we have separately looked into whether the PI name specifically, is compulsory.

**(iii) The audit trail of each record:** We wished to know whether, where relevant, a given trial (a) has an audit trail, and if so, (b) whether the changes are clearly highlighted; and (c) whether two versions of the record can be readily compared. In some cases where the sample trial, used for most analyses, did not have a history of changes, we used another sample trial, whose URL is provided in S5 Table.

**(iv) The flagging of retrospectively registered trials:** We documented whether each PR + specifically mentions the registration status of the trial (prospective vs retrospective), or flags retrospectively registered trials.

**(v) The reason for the termination of a trial, if applicable:** In this case, we first determined whether the PR+ have a category of terminated trials. For those that do, we captured whether or not a reason for trial termination is provided.

**III. Quality and validity, technical capacity, and data interchange standards features. (i) Use of a controlled vocabulary for the health condition category:** We evaluated whether (a) there is a drop down menu for choosing a term from a controlled vocabulary, (b) the registry recommends a widely used controlled vocabulary, or (c) the trialist has to use a free text box.

**(ii) The availability of documentation for the processes of the registry, or information on the site:** We evaluated the presence of three types of documents, that is (a) a glossary or the definition of each field of the record, (b) a list of frequently asked questions (FAQs) and (c) one or more user guides.

**(iii) Security features of the registry website:** The websites were checked for the presence of a basic security feature, an SSL certificate, as reflected in an 'https' in the website URL, instead of an 'http'.

---

Box 1. Terminology used in the study

Here, we list a few terms that have been used throughout the study, along with a description of what these refer to:

a  the word 'section' only refers to one or more of the nine sections of the ISCTR;

b  the word 'feature' only refers to one or more of the 14 features of each registry that are the focus of this study; and

c  since the different registries may have different variants of each 'feature', we use the word 'variation' in this context.

---

## The scorecard

Overall, 14 features of the PR+ were assessed. Each registry has a particular variant of a given feature, which may be more useful or less so. A scoring rationale was devised for every feature analysed, based on which each registry received a score for its variation of a given feature. The rationale is described in detail in Table 1, and further details are provided in the Results and Discussion section.

The following general rules were applied for the scoring system. These are illustrated by particular features in Table 1.

i. If the feature is absent, the registry gets a score of 0. This is illustrated in features 1.2 and 1.5.

ii. For features with multiple variations, the score ranges from 1 to 5 based on pre-set criteria, as defined in Table 1. This is illustrated in features 1.1 and 2.1.

iii. For certain features, which involve counts of fields present, the score increases by 1 point per field. This is illustrated in features 1.3 and 2.2.

iv. In case a registry has multiple possible scores for a particular feature, the highest one is awarded. This is illustrated in feature 1.5.

## Results and discussion

We first documented basic information about each of the registries. The full name of each registry, its acronym, the country where it is based, and the year it was established are provided in Fig 1 and Table 2. Except CTG, the acronyms used for each registry are the official acronyms. All but one of the PR+ were established between 2000 and 2010, inclusive. LBCTR was established in 2019. Eight registries (ANZCTR, ChiCTR, CTG, DRKS, IRCT, ISRCTN, JPRN, and SLCTR) allow trial registrations from all countries, and the rest usually from the country where the registry is based, or from specific countries or regions. For example, PACTR caters to clinical trials conducted in Africa. On 18 April 2020, the registries cumulatively held 572,901 records, with CTG accounting for 336,444 (59%).

Trials may be registered either prospectively or retrospectively, that is before the enrolment of the first participant or after. Six of the PR+ (CTRI, IRCT, LBCTR, REPEC, SLCTR and TCTR) only allow prospective registration, whereas the remaining accept retrospective as well. Five of the PR+ (EU-CTR, IRCT, PACTR, REPEC and SLCTR) accept only interventional clinical trials, while the remaining may accept others such as observational studies, post marketing surveys or expanded access programs. All the registries use English, and 11 of them display some or all information in another language as well.

We then analysed 14 features of the PR+, which have been grouped according to the sections of ISCTR that they map to (S1 Table). In Table 3, we list the score obtained by each PR + per feature, and overall. We also provide the maximum score possible per feature. Further details are provided below, or are available in relevant Supplementary files, which are referenced in Table 1.

**1. Accessibility.** One of the principal reasons for the existence of clinical trial registries is to provide the public with information, and to thereby increase trust in the trial enterprise [28]. Therefore, we first examined the accessibility of information in the PR+. For this, we assessed several features, as described below:

**(i) Ease of obtaining the total number of trials:** As a first step, it is important to know how many records the database holds. This number should be readily available, and we have therefore analysed the ease of accessing it. The five registries (CRIS, CTG, EU-CTR, IRCT and

**Table 1. Rationale for score given to each registry for features used to create the Scorecard.**

| Feature analyzed | | Rating scale and rationale | Relevant Supplementary file |
|---|---|---|---|
| **1** | **Accessibility** | | |
| 1.1 | Total number of trials in the registry | Number displayed on home page: 5 | S2 Table |
| | | Number available after a search: 3 | |
| | | Number needs to be calculated: 2 | |
| | | Discrepant information at different places on the site: 1 | |
| 1.2 | Existence of Basic search function | Presence of a basic search function: 5 | S2 Table |
| | | Absence of a basic search function: 0 | |
| 1.3 | Advanced search function–TRDS fields | Each TRDS field: 1 | S2 Table |
| 1.4 | Advanced search function–Extra fields | Each extra field: 1, but with a cap of 5 overall, because of the idiosyncratic nature of some of the search possibilities. | S2 Table |
| 1.5 | Data download options | Excel/csv/tsv: 5 | S2 Table |
| | | HTML/XML: 2 | |
| | | Word/txt/pdf: 1 | |
| | | No download options: 0 | |
| | | Since all the registries except NTR permit HTML downloads (even if it is not explicitly stated), no registry gets a rating of '1'. | |
| **2** | **Content or TRDS sections** | | |
| 2.1 | Brief view: TRDS fields | 10 or more fields, which are customizable, and wrapping of text: 5 | S3 Table |
| | | 10 or more fields, which are customizable, but without wrapping of text: 4 | |
| | | A fixed number of fields, that are more than 3: 3 | |
| | | Up to 3 fields: 1 | |
| 2.2 | Brief view: Extra fields | Each field: 1 point | S4 Table |
| 2.3 | Detailed view: TRDS fields | The number of fields over 20 | S3 Table |
| 2.4 | Detailed view: Extra fields | Each field: 1 point | S4 Table |
| | | In this case, the maximum score is dictated by the registry with the maximum number of fields. | |
| 2.5 | Whether PI name is compulsory | PI name is compulsory: 5 | S5 Table |
| | | It is not clear whether the scientific contact is the PI (regardless of whether or not this information is compulsory): 2 | |
| | | There is a field for the PI name, but it is not clear whether the information is compulsory: 2 | |
| | | The PI name is voluntary: 0 | |
| 2.6 | Audit trail | Each of the following aspects receives 1 point: (i) the existence of an audit trail; (ii) the changes made are clearly highlighted; and (iii) it is possible to compare any two versions of the record. | S5 Table |
| **3** | **Other Sections** | | |
| 3.1 | Health condition | A drop-down menu for choosing a term from a controlled vocabulary: 5 | S5 Table |
| | | A widely used controlled vocabulary is recommended: 3 | |
| | | Free text field: 1 | |
| 3.2 | SSL certificate | Website secured with SSL: 3 | S5 Table |
| | | Website not secured with SSL: 0 | |
| 3.3 | Documentation | Provides (a) a glossary or the definition of each field of the record; (b) List of FAQs; (c) One or more user guides: 1 point each. No points are awarded for the quality of these documents. | S5 Table |

The relevant Supplementary files with further details are also referenced.

ReBEC) that list it on the homepage were given the highest score of 5. Ten registries display this number after a search for all trials, and received a score of 3. Two registries (RPCEC, SLCTR), for which the number of records is available only by a manual calculation, received 2. LBCTR provides discrepant information at different places on the site, and thus received the

## A

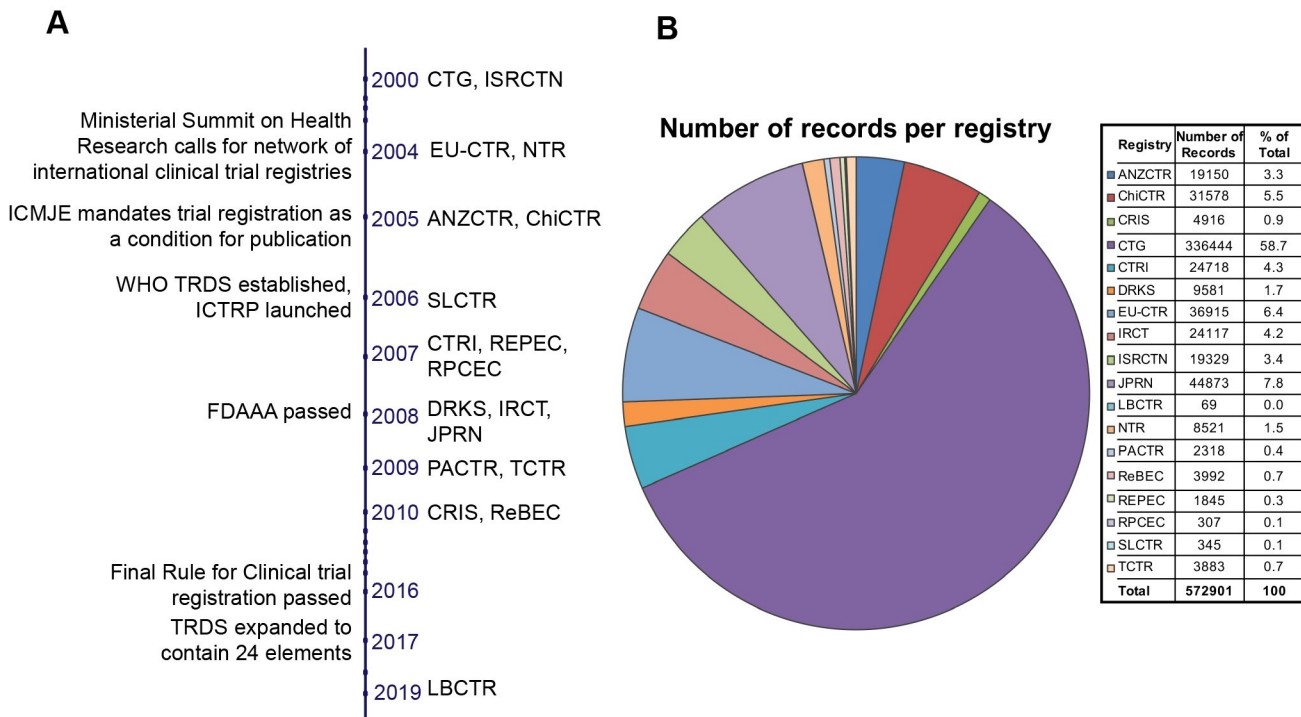

**Timeline A:**

| Year | Event / Registries |
|---|---|
| | Ministerial Summit on Health Research calls for network of international clinical trial registries |
| | ICMJE mandates trial registration as a condition for publication |
| | WHO TRDS established, ICTRP launched |
| | FDAAA passed |
| | Final Rule for Clinical trial registration passed |
| | TRDS expanded to contain 24 elements |

| Year | Registries |
|---|---|
| 2000 | CTG, ISRCTN |
| 2004 | EU-CTR, NTR |
| 2005 | ANZCTR, ChiCTR |
| 2006 | SLCTR |
| 2007 | CTRI, REPEC, RPCEC |
| 2008 | DRKS, IRCT, JPRN |
| 2009 | PACTR, TCTR |
| 2010 | CRIS, ReBEC |
| 2016 | |
| 2017 | |
| 2019 | LBCTR |

## B

**Number of records per registry**

| Registry | Number of Records | % of Total |
|---|---|---|
| ANZCTR | 19150 | 3.3 |
| ChiCTR | 31578 | 5.5 |
| CRIS | 4916 | 0.9 |
| CTG | 336444 | 58.7 |
| CTRI | 24718 | 4.3 |
| DRKS | 9581 | 1.7 |
| EU-CTR | 36915 | 6.4 |
| IRCT | 24117 | 4.2 |
| ISRCTN | 19329 | 3.4 |
| JPRN | 44873 | 7.8 |
| LBCTR | 69 | 0.0 |
| NTR | 8521 | 1.5 |
| PACTR | 2318 | 0.4 |
| ReBEC | 3992 | 0.7 |
| REPEC | 1845 | 0.3 |
| RPCEC | 307 | 0.1 |
| SLCTR | 345 | 0.1 |
| TCTR | 3883 | 0.7 |
| **Total** | **572901** | **100** |

**Fig 1.** A. The timeline of establishment of the PR+. Key events related to trial registration are also noted. B. Number of records per registry as on 18 April 2020. The pie chart shows the distribution of the number of records in each registry. The actual number, and as a percentage of the total, are also provided.

lowest score of 1. The median score obtained was 3. It is a trivial task to put the figure for the total number of trials on the home page, and we encourage all registries to do so.

For a significant fraction of users, the search functions are crucially important to access the information in a registry. ISCTR recommends that at the minimum, there must be a basic text search, and it must be possible to conduct searches within the interventions and conditions fields. Several PR+ go much further than this, and therefore we have conducted a detailed assessment of their search capabilities.

**(ii) Basic search function:** We determined the presence of a basic search function and have awarded a score of 5 to the 15 registries that provide it. Only three (ChiCTR, LBCTR and SLCTR) do not have this feature, and received 0. The median score was 5.

Most PR+ have a basic search function that enables search by keywords. This is a crucial aspect of the functionality of the trial registry website, and significantly increases the ease of searching for information and improves user experience.

**(iii) Advanced search function—TRDS fields:** We then examined how many of the 24 TRDS fields could be used in the Advanced search function. Out of a possible score of 24, where the registries received 1 point per field, the maximum score of 17 was attained by ChiCTR and IRCT. JPRN and NTR do not allow a search by any TRDS field and received 0. The remaining registries received scores between 1 and 15. The median score was 8.5.

**(iv) Advanced search function—Extra fields:** A few registries list fields other than the TRDS fields as part of the search function. Six PR+ (ChiCTR, CRIS, CTG, DRKS, IRCT and ISRCTN) have five or more Extra fields, and therefore received a score of 5. Six registries (ANZCTR, CTRI, EU-CTR, PACTR, ReBEC and REPEC) received scores ranging from 1–4. Six registries do not allow a search using any Extra fields, and received 0. The median score was 2.

**Table 2. An overview of each registry, listing its acronym, full name, country where it is based, countries from where registration is accepted, type of registration allowed, type of study hosted, and language used.**

| Registry acronym | Registry full name | Country where registry is based | Countries from where registration is accepted | Type of registration allowed | Type of study | Additional language[1] |
|---|---|---|---|---|---|---|
| ANZCTR | Australian New Zealand Clinical Trials Registry | Australia | All countries. However, trials in Australia and New Zealand are prioritized | Prospective, Retrospective[2] | Interventional, Observational | – |
| ChiCTR | Chinese Clinical Trial Register | China | All countries | Prospective, Retrospective | Interventional, Observational, Others | Chinese |
| CRIS | Clinical Research Information Service | Republic of Korea | Republic of Korea | Prospective, Retrospective | Interventional, Observational | Korean |
| CTG[3] | ClinicalTrials.gov | USA | All countries | Prospective Retrospective | Interventional, Observational, Expanded Access | – |
| CTRI | Clinical Trials Registry—India | India | Other countries in the region which do not have a Primary Registry of their own[4] | Prospective | Interventional, Observational, PMS[5], BA/BE[5] | – |
| DRKS | German Clinical Trials Register | Germany | All countries | Prospective, Retrospective | Interventional, Observational, Epidemiological, Others | German |
| EU-CTR | EU Clinical Trials Register (EU-CTR) | The Netherlands | All interventional trials that have at least one centre in the EU and EEA. Certain trials conducted entirely outside these regions. | Prospective. Retrospective if permitted by National Competent Authority of the Member State | Interventional | Older trials may have content in the host country's language |
| IRCT | Iranian Registry of Clinical Trials | Iran | All countries | Prospective | Interventional | Persian |
| ISRCTN | International Standard Registered Clinical/soCial sTudy Number | UK | All countries | Prospective, Retrospective[2] | Interventional, Observational | – |
| JPRN | Japan Primary Registries Network[6] | Japan | All countries | Prospective, Retrospective | Interventional, Observational | Japanese |
| LBCTR | Lebanon Clinical Trials Registry | Lebanon | Lebanon | Prospective | Interventional, Observational | Brief summary of the study is also available in Arabic |
| NTR | Netherlands Trial Register | The Netherlands | Trials conducted in Netherlands or involving Dutch researchers. | Prospective, Ongoing studies | Interventional, Observational | Some information may be available in Dutch |
| PACTR | Pan African Clinical Trials Registry | South Africa | All countries in Africa | Prospective, Retrospective | Interventional | – |
| ReBEC | Brazilian Registry of Clinical Trials | Brazil | Brazil[4] | Prospective Retrospective | Interventional, Observational | Portugese and Spanish, for some records, and in a limited way |
| REPEC | Peruvian Clinical Trial Registry | Peru | Peru | Prospective | Interventional | Spanish |
| RPCEC | Cuban Public Registry of Clinical Trials | Cuba | Cuba[7] | Prospective, Retrospective | Interventional Observational | Spanish |
| SLCTR | Sri Lanka Clinical Trials Registry | Sri Lanka | All countries | Prospective | Interventional | – |

*(Continued)*

**Table 2.** (Continued)

| Registry acronym | Registry full name | Country where registry is based | Countries from where registration is accepted | Type of registration allowed | Type of study | Additional language[1] |
|---|---|---|---|---|---|---|
| TCTR | Thai Clinical Trials Registry | Thailand | Thailand | Prospective | Interventional, Observational | – |

1. All registries are required to be in English. However, some provide content in additional language(s).

2. Retrospective registration is allowed but prospective registration is preferred and encouraged.

3. Except CTG, all the acronyms listed are the official acronyms.

4. For two registries (CTRI, REBEC) the information on the ICTRP portal and on their own websites is discrepant. Upon inspection, the latter sources appear to be correct, and we have described the registries accordingly.

5. PMS: post-marketing surveillance; BA/BE: Bioavailability/Bioequivalence.

6. Common forum for trials from three Japanese registries, that is (UMIN Clinical Trials Registry (UMIN-CTR), Japan Pharmaceutical Information Center Clinical Trials Information (JAPIC-CTI), and Japan Medical Association—Center for Clinical Trials (JMACCT)).

7. Trials are accepted from Cuban sponsors, conducting trials in Cuba or abroad, with Cuban or foreign products.

Notably, some registries (ChiCTR, EU-CTR, ISRCTN, PACTR, REPEC) were built on earlier versions.

Overall, the PR+ provided more fields in the Advanced search function than the minimum recommended by the ISCTR. This becomes especially relevant for researchers conducting systematic reviews, work which requires extensive searches to gather information on clinical trials in specific areas.

**(v) Data download options:** Having conducted a search, users may wish to download many fields of data, for many records. We therefore gave the highest score of 5 to the five registries (ANZCTR, CTG, DRKS, ISRCTN and JPRN) that allow data downloads in a csv, excel or tsv format. 12 registries provide HTML and XML options, and received a score of 2. Only NTR lacks any options for data download, and received 0. The median score was 2.

All the available data download options are adequate for the inspection of a few records, but it is essential that the PR+ provide bulk data download options such as csv, especially as an increasing number of users are shifting towards automated systems of analysis.

**2. Content and TRDS sections.** Next, we examined multiple features that map to the Content or TRDS sections, which overlap since the TRDS fields are a form of content. Below, we describe our scoring of the Brief and Detailed views of the PR+.

**(i) Brief view: TRDS fields:** Since the Brief view is primarily designed to provide an overview of the trial, it can be very helpful for a user if the number of fields in the Brief view can be customized. Therefore, we have given higher scores to registries that provide this option. Two registries (CTG and TCTR), display more than 10 TRDS fields, and allow customization and text wrapping. They received the maximum score of 5. CRIS displays more than 10 TRDS fields, that are customizable but without text wrap, and received 4. Eleven registries display more than three fields, which are fixed, and got a score of 3. The four registries (ISRCTN, NTR, RPCEC and SLCTR) that display three fields or less received 1. The median score was 3.

A customizable brief view of search results is extremely useful in a trial registry, where different types of users such as patients, healthcare professionals or sponsors, may be interested in different fields.

**(ii) Detailed view: TRDS fields:** The Detailed view tends to have all 24 TRDS fields. However, we found that all the PR+ do not yet list the four fields that have been included in the latest version of TRDS [38]. Eight registries (ANZCTR, ChiCTR, CTG, ISRCTN, JPRN, LBCTR, PACTR and SLCTR) do so, and received the highest score of 4. Most of the remaining PR+ display between one and three of the new fields and were scored accordingly. Only one registry,

**Table 3. The scorecard.**

| | | Max score | ANZCTR | ChiCTR | CRIS | CTG | CTRI | DRKS | EU-CTR | IRCT | ISRCTN | JPRN | LBCTR | NTR | PACTR | ReBEC | REPEC | RPCEC | SLCTR | TCTR |
|---|---|---|---|---|---|---|---|---|---|---|---|---|---|---|---|---|---|---|---|---|
| **1** | **Accessibility section** | | | | | | | | | | | | | | | | | | | |
| 1.1 | Total number of trials in the registry | **5** | 3 | 3 | 5 | 5 | 3 | 3 | 5 | 5 | 3 | 3 | 1 | 3 | 3 | 5 | 3 | 2 | 2 | 3 |
| 1.2 | Existence of Basic search function | **5** | 5 | 0 | 5 | 5 | 5 | 5 | 5 | 5 | 5 | 5 | 0 | 5 | 5 | 5 | 5 | 5 | 0 | 5 |
| 1.3 | Advanced search function–TRDS fields | **24** | 11 | 17 | 14 | 15 | 10 | 8 | 7 | 17 | 13 | 0 | 9 | 0 | 12 | 4 | 1 | 2 | 2 | 7 |
| 1.4 | Advanced search function–Extra fields | **5** | 1 | 5 | 5 | 5 | 4 | 5 | 3 | 5 | 5 | 0 | 0 | 0 | 2 | 1 | 2 | 0 | 0 | 0 |
| 1.5 | Data download options | **5** | 5 | 2 | 2 | 5 | 2 | 5 | 2 | 2 | 5 | 5 | 2 | 0 | 2 | 2 | 2 | 2 | 2 | 2 |
| | **SUB-TOTAL** | **44** | **25** | **27** | **31** | **35** | **24** | **26** | **22** | **34** | **31** | **13** | **12** | **8** | **24** | **17** | **13** | **14** | **6** | **17** |
| **2** | **Content or TRDS sections** | | | | | | | | | | | | | | | | | | | |
| 2.1 | Brief view: TRDS fields | **5** | 3 | 3 | 4 | 5 | 3 | 3 | 3 | 3 | 1 | 3 | 3 | 1 | 3 | 3 | 3 | 1 | 1 | 5 |
| 2.2 | Brief view: Extra fields | **5** | 3 | 1 | 2 | 2 | 0 | 3 | 1 | 3 | 1 | 1 | 1 | 0 | 3 | 0 | 0 | 1 | 5 | 1 |
| 2.3 | Detailed view: TRDS fields | **4** | 4 | 4 | 3 | 4 | 3 | 2 | 2 | 2 | 4 | 4 | 4 | 3 | 4 | 0 | 3 | 1 | 4 | 3 |
| 2.4 | Detailed view: Extra fields | **15** | 10 | 5 | 10 | 15 | 8 | 6 | 9 | 7 | 10 | 0 | 9 | 5 | 5 | 4 | 10 | 6 | 5 | 9 |
| 2.5 | Whether PI name is compulsory | **5** | 5 | 5 | 5 | 0 | 0 | 5 | 5 | 2 | 2 | 2 | 2 | 2 | 5 | 0 | 2 | 5 | 2 | 2 |
| 2.6 | Audit trail | **3** | 1 | 1 | 3 | 3 | 1 | 3 | 0 | 3 | 2 | 0 | 1 | 0 | 3 | 0 | 0 | 3 | 2 | 0 |
| | **SUB-TOTAL** | **37** | **26** | **19** | **27** | **29** | **15** | **22** | **20** | **20** | **20** | **10** | **20** | **11** | **23** | **7** | **18** | **17** | **19** | **20** |
| **3** | **Other sections** | | | | | | | | | | | | | | | | | | | |
| 3.1 | Health condition | **5** | 5 | 3 | 5 | 3 | 5 | 5 | 5 | 3 | 1 | 5 | 5 | 1 | 5 | 5 | 3 | 1 | 1 | 3 |
| 3.2 | SSL certificate | **5** | 5 | 0 | 5 | 5 | 0 | 5 | 5 | 5 | 0 | 5 | 0 | 5 | 5 | 0 | 5 | 0 | 5 | 5 |
| 3.3 | Documentation | **3** | 3 | 2 | 2 | 3 | 3 | 3 | 3 | 1 | 3 | 0 | 3 | 0 | 2 | 3 | 2 | 2 | 2 | 3 |
| | **SUB-TOTAL** | **13** | **13** | **5** | **12** | **11** | **8** | **13** | **13** | **9** | **4** | **10** | **8** | **6** | **12** | **8** | **10** | **3** | **8** | **11** |
| | **TOTAL** | **94** | **64** | **51** | **70** | **75** | **47** | **61** | **55** | **63** | **55** | **33** | **40** | **25** | **59** | **32** | **41** | **34** | **33** | **48** |
| | **% of TOTAL** | | **68** | **54** | **74** | **80** | **50** | **65** | **59** | **67** | **59** | **35** | **43** | **27** | **63** | **34** | **44** | **36** | **35** | **51** |
| | **Rank of each registry** | | **3** | **9** | **2** | **1** | **11** | **5** | **7** | **4** | **7** | **15** | **13** | **18** | **6** | **17** | **12** | **14** | **15** | **10** |

The list of features used to create the Scorecard; the maximum score per feature; the score obtained by each registry per feature, and overall per section; the total score per registry; and the rank of each registry.

ReBEC, has not been updated to display any of the new fields, and received a score of 0. The median score was 3.

We hope that over time more registries will be in full compliance with the ISCTR-mandated fields.

**(iii) Extra fields:** Registries list Extra fields in both the Brief and Detailed views. In the Brief View, only SLCTR received the maximum score of 5. Four registries (CTRI, NTR, ReBEC and REPEC) have no Extra fields and received a score of 0. The remaining have between one and three of such fields and were scored accordingly. The median score was 1.

In the Detailed View, most registries have between five and 10 Extra fields. However CTG has 15, and JPRN has none. The median score was 7.5.

Some of the Extra fields, such as the date of last update, and whether registration was prospective or retrospective, are recommended by ISCTR. Interestingly, one-third or more of the registries list several fields that the ISCTR does not specifically recommend. This seems to reflect a certain level of agreement among the managers of registries that particular fields are important. There may be a range of reasons for including these fields. For instance, India had been criticized for the lack of appropriate oversight to ensure the ethical conduct of trials, and therefore CTRI asked trialists for details of the ethics committee even before ISCTR required this information [39, 40]. Also, there have been demands from the Cochrane collaboration, and many other individuals and groups, to include several additional items in the ISCTR list, which WHO has not agreed to. It is alleged that the recommended list is closer to what industry demanded [41]. As such, although ISCTR may not list every field that many people believed to be essential, managers of particular registries may have chosen to list some of them.

We explored some of these Extra fields in greater detail below.

**(iv) Whether PI name is compulsory:** For the sake of accountability it is important that the field 'PI name' is compulsory [25, 38]. Although we have assessed Contact for Scientific Queries as a TRDS field, we have not assumed that this person is the PI, and therefore have separately looked into whether the PI name is compulsory. In seven registries (ANZCTR, ChiCTR, CRIS, DRKS, EU-CTR, PACTR and RPCEC) it is so, and they received the highest score of 5. Several have either not made it clear whether the scientific contact is the PI, or have a separate field for the PI name but have not stated whether it is compulsory. They each received a score of 2. Three registries (CTG, CTRI and ReBEC) have marked this field as voluntary, and received 0. The median score was 2.

WHO documents [25, 38] have contradictory information on the issue of PI name and Contact for Scientific Queries. They require that the PI's name, title and email ID be provided, but state that this should be a functional name, not a personal one. ISCTR states that the PI is the Contact for Scientific Queries, unless the PI delegates this task to somebody else. If the PI name is compulsory–and preferably recorded in a fixed format [42]–then this information will enable researchers to quantify the number of unique PIs in a country, ask whether a PI has been taking on too many trials, and perform other analyses. Therefore we commend the registries that have made this field compulsory.

**(v) Audit trail:** ISCTR requires that the audit trail of each record should be publicly available and so we have examined the presence and usefulness of this feature. Six registries (CRIS, CTG, DRKS, IRCT, PACTR and RPCEC) have the option of comparing two versions of the trial record and received the maximum score of 3; two (ISRCTN and SLCTR) have highlighted the changes made to a trial record, and got 2; four (ANZCTR, ChiCTR, CTRI and LBCTR) have a basic form of an audit trail and got 1; and six of the PR+ do not provide any audit trail and got 0. The median score was 1. It is clear that most registries do not have an ideal audit trail.

The information pertaining to the following two features is present in the Extra fields, either as a separate field in the Detailed view, or marked with a flag in the Brief View. Hence we have not scored these features separately.

**(vi) Flagging retrospective or prospective registration status of a trial:** Prospective registration is crucial to prevent unrecorded 'outcome switching', which creates a bias in the medical evidence base [28]. Nevertheless, it has been argued that (i) it is a duty to trial participants to register each trial, and subsequently publish the results, and (ii) not registering a trial could lead to its loss from the documented universe of trials [43].

As such, retrospective registration is better than non-registration, and therefore many PR + permit it. We have documented this in Table 1.

Users may have more confidence in the results of a prospectively than a retrospectively registered trial. Further, flagging retrospectively registered ones may shame the registrants into registering prospectively in future [14]. Accordingly, we have analysed whether PR+ highlight the registration status of trials and flag retrospectively registered ones.

Over half of the PR+ do so.

**(vii) The reason for the termination of a trial, if applicable:** It is important to know why a study was terminated as it provides economic, ethical and scientific insights that can help improve ongoing or upcoming clinical trials [44]. Our analysis showed that only eight registries provide this information at all, and only three provide drop-down menus of reasons for termination (S5 Table). Researchers who have studied the leading causes of trial termination have suggested that the cause should be selected from a fixed set of options [45].

**3. Other sections.** Finally, we examined three features that map to other sections of the ISCTR, as follows: (a) Health condition, (b) the presence of a Secure Sockets Layer (SSL) certificate and (c) Documentation.

**(i) Health condition:** First, the issue of classifying health conditions, which maps to Data Interchange Standards. We find that only half the PR+ provide drop-down menus for this field, and they received the highest score of 5. Five registries (ChiCTR, CTG, IRCT, REPEC and TCTR) recommend the use of standardized vocabulary, and received 3. Four registries (ISRCTN, NTR, RPCEC and SLCTR) that do not provide such options, and have a free text field for health condition, received 1. The median score was 4.

Comparisons across registries are easier if each one uses a controlled vocabulary, and in particular one that maps to a widely-used metathesaurus [46] as recommended in ISCTR [25]. It is therefore preferable that the health condition be selected from a fixed set of options.

**(ii) The presence of an SSL certificate:** Second, the security of the website. In the Technical Capacity section, ISCTR requires that each registry have adequate protection against the corruption or loss of data. We have assessed something basic, that is whether the website is secured with an SSL certificate, as is evident when a website URL contains 'https'. We find that only 12 of the PR+ websites have this certification, and each received a score of 5. The remaining six registries (ChiCTR, CTRI, ISRCTN, LBCTR, ReBEC and RPCEC) have URLs with an 'http', and received 0. The median score was 5.

The SSL certificate is an important tool to safeguard data of the registry and that of its users, and Google currently marks all sites without it as insecure [47]. As such, it is also a sign of credibility for a user who may hesitate to access a site that lacks a security certificate.

**(iii) Documentation:** Third, the issue of documentation. Various documents help users to understand the processes of a registry, or the data it hosts. Only half the PR+ provide all the three types of documentation we have assessed, and received a score of 3. Six registries (ChiCTR, CRIS, PACTR, REPEC, RPCEC and SLCTR) have only two types of documents and received 2, and one registry (IRCT) displays only a user guide and received a score of 1. Two

registries (JPRN and NTR) do not provide any documentation, and have received 0. The median score obtained was 2.5.

Although the three documents that we have scored are not explicit requirements of the ISCTR, they assist users in registering their trial correctly. As such, this feature maps to the Quality and Validity section.

In general, we have barely touched upon Quality and Validity, since investigating the completeness or quality of the records in the PR+ would be a large exercise in itself. The sponsor needs to ensure a high quality trial record, but this may not happen, and various studies have highlighted deficiencies in the records of different registries [10–13, 48]. It is also the duty of the managers of the registry to facilitate better quality trial registration, as has been recommended by ISCTR. Additionally, for several of the minimum standards recommended by ISCTR, either it is not possible for us to assess compliance, or the requirements do not immediately impact use of the registry data. Therefore we have also barely touched upon Unambiguous Identification (although Secondary identifying numbers, a field in TRDS, also maps to this section), Technical Capacity, and Data Interchange Standards. Further, we have not touched upon the sections (i) Administration and Governance, and (ii) Partner Registries.

Overall, as derived from an assessment of 14 features described above, the maximum score that any registry obtained was 94 points (Table 3). The PR+ received scores ranging from 27% (NTR) to 80% (CTG) of the maximum, with an average of 52%. Despite the limited nature of our audit, the lowest- and highest-scoring registries received scores that differ by over 50%. To the best of our knowledge, this widely divergent quality of the PR+ has not been documented before.

## An ideal registry

We found that the registries show a high degree of variability for a given feature, ranging from a sophisticated to a routine variation, or its complete absence. We have used the best variant of the features analyzed to define an interim ideal registry. In such a registry,

i. the total number of trials is displayed on the home page;

ii. a search is possible through (a) a basic search function, (b) each of the TRDS fields, and (c) a few extra fields;

iii. the data download options include a csv, excel, or tsv format, and support automated bulk downloads;

iv. the Brief view is customizable, with 10 or more fields, with text wrapping;

v. the Detailed view includes all the TRDS fields;

vi. there is clarity on whether or not the scientific contact is the PI;

vii. the PI name is compulsory;

viii. the reason for the termination of a trial is selected from a drop-down menu of possible reasons;

ix. each trial has an audit trail that enables a comparison of any two versions;

x. at the very least, the following documents are provided, in English: (a) a definition of each field of the record, (b) a list of FAQs, and (c) one or more user guides;

xi. the website is secured with an SSL certificate; and

xii. the health condition category is chosen from a drop-down menu with a controlled vocabulary, preferably a widely used one.

The ISCTR recommends several other standards including higher data quality, more complete records and the reporting of results. Although it is hoped that all registries will implement all of these standards in due course, in the interim, registries may wish to implement the list above if they have not already done so.

Registries have many users. The scorecard above analyses features that are of interest to the authors and, by extension, possibly to other researchers concerned with the health of the trial ecosystem overall. Other categories of users, such as medical professionals, patients, trial sponsors, policy makers, data scientists and so on, may wish to alter the assessed features, or the scoring, in order to rank the registries according to their priorities. For instance, a data scientist would be very appreciative of ANZCTR, which specifically enables web crawling of its records [49]. Furthermore, the managers of other registries, either public or private, and either based on the data in the PR+ or not, may be interested in the results of this study.

The ongoing Covid-19 pandemic has forcefully brought home the need for high quality trial registries with information that is consistent, comprehensive and available in a user-friendly fashion. Billions of people need to be immediately protected from the virus, and large numbers of drugs and vaccines are in trials. There is world-wide interest in these trials, and information that is being tracked includes what is being trialled; where these trials are taking place; and the results of these trials. Each country needs to take public health decisions, which will evolve as the evidence from trials running in different parts of the world yield results. Public trial registries are one of the fastest ways of communicating these results.

Further, the publicly available, freely accessible information in such registries helps to build trust with the public [26, 44]. Covid-19 trials have been among the fastest recruiting ones in history [50, 51], and it is possible that the publicly available information in trial registries has helped many of the potential trial participants decide to enrol.

It is not just that everyone is interested in the positive outcomes of trials. For example, an inspection of the CTRI records of hundreds of Covid-19 trials being run in India has thrown up quality issues in almost all of them. Based on negative publicity, the government has taken action in some cases [52].

A handful of the registries evaluated in this study have provided a customized search for Covid-19 trials on their home pages. Covid-19 knowledge hubs, that facilitate access to trials that are registered with existing registries such as those described in this work, have also been set up [53–55]. Undoubtedly, such efforts to catalogue all Covid-19 trials, and experiences with Covid-19, has contributed to a more rapid understanding of the virus, and a more efficient management of the pandemic.

This makes the case for (a) more of such knowledge hubs, or (b) a single world-wide trial registry. Arguments for a single world-wide registry have been made for many years [56] However there are also counter-arguments, which highlight the fact that local registries (a) meet local ethics and regulatory requirements; (b) are easier to establish and monitor; (c) can be customized to local requirements; (d) are dependent on local infrastructure; (d) provide the national trial landscape and (e) can be a source of national pride [56, 57].

Given that there are pros and cons to having a single world-wide registry, the best way forward may be to work to improve PR+ registries. If higher data quality and data sharing standards were more widespread, this would facilitate interoperability and data transfer while maintaining data integrity.

There is a long history of various stakeholders arguing for the need to improve registries and the quality of trial registration. Examples include academics and health activists [58–60], journals (ICMJE) [61], WHO [41], registry managers [62], funders [63, 64], and governments [65]. Each of these efforts has led to some improvements in the number and quality of trial records hosted by registries. However none of them has led to a perfect set of records. It is likely

that the only way this will be achieved is if all stakeholders continue to apply pressure on the registries. Studies such as this one help to highlight deficiencies, which adds to the other efforts aimed at improving registries. The authors would welcome other researchers' efforts to create and update a website that lists the scorecard, with periodic updates. Should such a website not be created by any other group, the authors intend to re-evaluate the registries' performance on the scorecard every few years.

In summary, to the best of our knowledge, this is the first study undertaking a comparative analysis of WHO-recognized registries to assess compliance to ISCTR. Our use of a scorecard, based on pre-set criteria, ensured an impartial quantification of the quality of the features analyzed across the PR+. As such, even though our study analyzed a limited set of features, it clearly shows the substantial variation in compliance with the recommended minimal standards. Our study would be helpful to researchers who may wish to extend this audit and evaluate the completeness of the records or the quality of their data, two other major issues, in all 18 registries.

This study has a few limitations, as follows: (i) It assesses only some of the many features in each registry. In particular, it does not evaluate any aspect of trial methodology or results, which are crucial portions of such registries. As such, otherwise outstanding registries may have fared less well than expected. (ii) We have not evaluated the completeness of any records or the quality of their data. (iii) Each registry has been evaluated with respect to the list of fields in a recently registered trial. Earlier records in the same registry may have different content if the required details have changed over time. (iv) We have primarily focused on information that is available in English and may have missed important content in other languages. (v) Although applied systematically, the absolute values of the scores are arbitrary

## Conclusions

Over the years, CTG has received most of the attention of those interested in the accessibility and integrity of the data in public trial registries. As noted above, 41% of the records are held in the other PR+, and need to be examined as well. We have identified the best variations of several features that have already been implemented by one or more of these registries, and which serve as pointers on how the others may improve. Running a registry is not merely a bureaucratic task, but is part of a mission to safeguard patients' lives, and the ethics and science of medicine. We hope that our analysis is of some assistance in this.

## Supporting information

**S1 Table. Mapping to ISCTR.** The 14 features analyzed in this study map to the following nine sections of ISCTR: (i) Content, (ii) Quality and Validity, (iii) Accessibility, (iv) Unambiguous Identification, (v) Technical Capacity, (vi) Administration and Governance, (vii) The 24-field TRDS, (viii) Partner Registries, and (ix) Data Interchange Standards.
(XLSX)

**S2 Table. Data on six aspects of each registry.** (a) Total number of trials in the registry, (b) Existence of a basic search function, (c and d) Advanced search function–TRDS fields and Extra fields, and (e) Data download options.
(XLSX)

**S3 Table. The list of the TRDS fields that are present in the Brief view and the Detailed view.** The presence or absence of the field is indicated by a 1 or 0, respectively. The number of the sample trial used for each registry is also provided.
(XLS)

**S4 Table. For each registry, a listing of the Extra fields in the Brief and Detailed views.**
(XLSX)

**S5 Table. Data on multiple aspects of each registry.** (a) whether the PI name is compulsory; (b) reason for the termination of a trial, and whether there is a drop-down menu of reasons; (c) audit trail; (d) health condition (e) SSL certificate, and (f) documentation. (XLSX)

## Author Contributions

**Conceptualization:** Gayatri Saberwal.

**Data curation:** Nisha Venugopal.

**Formal analysis:** Nisha Venugopal, Gayatri Saberwal.

**Funding acquisition:** Gayatri Saberwal.

**Investigation:** Nisha Venugopal.

**Methodology:** Nisha Venugopal, Gayatri Saberwal.

**Project administration:** Gayatri Saberwal.

**Resources:** Gayatri Saberwal.

**Software:** Nisha Venugopal.

**Supervision:** Gayatri Saberwal.

**Validation:** Gayatri Saberwal.

**Visualization:** Nisha Venugopal, Gayatri Saberwal.

**Writing – original draft:** Gayatri Saberwal.

**Writing – review & editing:** Nisha Venugopal, Gayatri Saberwal.

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
