## [Decision Letter · Decision Letter 0]

27 Aug 2020

PONE-D-20-16268

A comparative analysis of important public clinical trial registries, and a proposal for an interim ideal one

PLOS ONE

Dear Dr. Saberwal,

Thank you for submitting your manuscript to PLOS ONE. After careful consideration, we feel that it has merit but does not fully meet PLOS ONE’s publication criteria as it currently stands. Therefore, we invite you to submit a revised version of the manuscript that addresses the points raised during the review process.

We look forward to receiving your revised manuscript.

Kind regards,

Dermot Cox

Academic Editor

PLOS ONE

Journal Requirements:

2. Thank you for including the following funding information; "

GS received internal institutional funds. These were partially from the Government of

Karnataka’s Department of Information Technology, Biotechnology and Science &

Technology (https://itbtst.karnataka.gov.in/english). There was no grant number.

Reviewers' comments:

Reviewer's Responses to Questions

**Comments to the Author**

1. Is the manuscript technically sound, and do the data support the conclusions?

Reviewer #1: Yes

Reviewer #2: No

2. Has the statistical analysis been performed appropriately and rigorously? 

Reviewer #1: Yes

Reviewer #2: N/A

3. Have the authors made all data underlying the findings in their manuscript fully available?

Reviewer #1: Yes

Reviewer #2: Yes

4. Is the manuscript presented in an intelligible fashion and written in standard English?

Reviewer #1: Yes

Reviewer #2: No

5. Review Comments to the Author

Reviewer #1: Overall this is a useful and very timely piece of work which, as the authors say, could trigger further work on registry assessment and a wider debate on how trial registries can both improve the features they offer and become more consistent. In general it is well written and well referenced, and is supported by a comprehensive set of detailed data as supplementary files. The authors acknowledge the limitations of their study and include a useful set of suggestions for an 'ideal registry' as an aspiration to work towards.

I have some reservations about some aspects of the paper, however, which I think detract from its overall quality – but which I hope can be easily rectified:

1) I found the organisation of some of the material confused. In particular the very short methods section provides little detail about the 17 features selected as the basis of assessment, how and why they were selected, and by whom, and how decisions on weighting were made, and why 3 were not assessed. Later on, in table 2b and as a large part of the 'Discussion of specific features…' in Box 1, much of this material is covered, but I think it would have been simpler and more logical to bring these explanations together as part of an expanded methods section. Box 1 is embedded in the discussion but its content seems largely a justification of the scorecard's construction. The result is that the reader has to work harder than they should to understand how and why the scoring system was constructed.

2) Similarly I think the results section could be better organised. Why not simply go through the results for each of the 14 areas assessed, noting at that point the median and the range of scores, techniques and difficulties in assessment, and possible caveats around the scores obtained? The current section provides useful tables and a brief summary, but much of the text is simply restating what was accessed. Would a simple pie chart be a useful way of summarising the total numbers data in table 1, to show the proportion of total registry entries included in each?

3) A minor point, but there 10 superscript references in Table 1, presumably to some explanatory notes about the data point presented, but I could not find any explanation for them, either in the main text or the supplementary material. They should either be removed or (better) the explanatory notes should be provided.

4) I thought the discussion was a little timid. The work was done in early 2020, in the context of a pandemic that has dramatically underscored the need for good quality, consistent and easily available information from trial registries, partly to be able to track the numbers, types and results of trials relating to COVID-19, partly because public health decisions require a network of data sources at a global level and registries should be a key part of this. That point might have been worth including – improving trial registry systems has become more urgent!

5) Similarly, although there is a general sentiment expressed that registries should improve, there were no concrete suggestions as to how this might be achieved or who needs to be involved, e.g. by greater collaboration between registries, perhaps orchestrated by the WHO, or by using the influence of funders and publishers to re-iterate the need for greater consistency. Are some of the aspects that were assessed easier to improve than others? If so how could they be progressed? Should there be a web page with a regularly updated 'score card' for the trial registries? I appreciate this was an initial survey but I think it might have been useful to venture, if only briefly, into this area in the discussion.

6) Another issue largely missing from the discussion: the authors mention that registries have many different types of users – researchers, clinicians, members of the public, data scientists, etc. I wonder if this should therefore lead to different scoring systems – perhaps with different weightings and / or items – for each of those major user groups. Those could provide additional insight into the strengths and weakness of different repositories, and thus more clearly identify areas of improvement, but could also be consolidated into an overall score if desired. For example, although the authors state their assumption is that most users would not have the technical expertise to use APIs, and / or scraping and crawling systems to retrieve data, the integration of trial registries with other data systems, and thus the ability to support bulk download by machines, is becoming increasingly important. I would have liked to have seen this aspect more explicitly included in any 'to do list' of possible future assessments, along with considerations of data quality, completeness, and the support for reporting results.

7) There is a minor but distracting typo in the first paragraph of the Results section (5,72,901)

Having listed all of the points above I would re-iterate that overall I think the paper is useful and should be published. The points are offered as suggestions for possible improvement.

Reviewer #2: The concept is interesting but needs to be re-written. The paper should first start with a good explanation of the origins of the ISCTR. For example, "following the Ministerial Summit on Health Research that took place in Mexico City, Mexico, in November 2004, participants called for the WHO to facilitate the establishment of: "a network of international clinical trials registers to ensure a single point of access and the unambiguous identification of trials". https://www.who.int/ictrp/about/en/ The authors need to be more complete in explaining the WHO registry network including primary vs partner registries as well as data providers; the differences of each. Then as it relates to registries what kind of papers have been published; findings; some of this is introduced at a high level in the discussion section which belongs in the introduction. The authors are not clear in their terminology (for example, versions vs features). There is reference to WHO's 24-field Trial Registration Data Set vs 17 features vs 14 features selected by the authors; there is reference to the 9 standards; hence it is not clear how these "concepts" inter-relate (24 vs 17 vs 9; data set vs standards) and why the authors selected 14 features (which is perhaps more attributes than features). How does WHO refer to author defined "features". The score card is hard to follow; the scoring is not understandable (for example, "for features with multiple variants, the score ranges from 1 to 5" is not clear and then how scores were calculated: as in the case of chCTR for advanced search fields TRDS a score of 17 was assigned). The meaning of the scorecard is not clear; the interpretation of the findings are inconsistent and leaves the reader bewildered. Please see specific comments in the attached word document. The topic is interesting but the paper needs to be rewritten and the concept of a scorecard has to be rethought to ensure it has logical relevance to the reader, that the scoring is understandable and can be interpreted thereby leading to actionable insights. The paper has to be placed in context of other relevant studies completed to date.

6. PLOS authors have the option to publish the peer review history of their article (what does this mean?). If published, this will include your full peer review and any attached files.

Reviewer #1: **Yes: **Steve Canham (Data Projects Manager, European Clinical Research Infrastructure Network, ECRIN)

Reviewer #2: No

---

## [Author Response · Author response to Decision Letter 0]

7 Oct 2020

Response to Reviewers

REVIEWER #1: Overall this is a useful and very timely piece of work which, as the authors say, could trigger further work on registry assessment and a wider debate on how trial registries can both improve the features they offer and become more consistent. In general it is well written and well referenced, and is supported by a comprehensive set of detailed data as supplementary files. The authors acknowledge the limitations of their study and include a useful set of suggestions for an 'ideal registry' as an aspiration to work towards.

I have some reservations about some aspects of the paper, however, which I think detract from its overall quality – but which I hope can be easily rectified:

Authors: We thank the reviewer for the appreciative comments. 

- - - - - - - - - - - - - - - - - -

1) I found the organisation of some of the material confused. In particular the very short methods section provides little detail about the 17 features selected as the basis of assessment, how and why they were selected, and by whom, and how decisions on weighting were made, and why 3 were not assessed. Later on, in table 2b and as a large part of the 'Discussion of specific features…' in Box 1, much of this material is covered, but I think it would have been simpler and more logical to bring these explanations together as part of an expanded methods section. Box 1 is embedded in the discussion but its content seems largely a justification of the scorecard's construction. The result is that the reader has to work harder than they should to understand how and why the scoring system was constructed.

Authors: We have reorganized the Methods section, and added further details. This includes portions from Box 1. We have also moved Table 2b to the methods section (where it is now Table 1) to make the rationale of the scorecard available upfront. Additionally, we wish to highlight that lines 128–130 describe how the authors selected the criteria for the scorecard, based on a review of the literature, but mainly focussing on the ISCTR guidelines.

- - - - - - - - - - - - - - - - - -

2) Similarly I think the results section could be better organised. Why not simply go through the results for each of the 14 areas assessed, noting at that point the median and the range of scores, techniques and difficulties in assessment, and possible caveats around the scores obtained? The current section provides useful tables and a brief summary, but much of the text is simply restating what was accessed. Would a simple pie chart be a useful way of summarising the total numbers data in table 1, to show the proportion of total registry entries included in each?

Authors: We have rewritten and reorganized the Results and Discussion. We have created the suggested pie chart, and have also presented other data from the erstwhile Table 1 as a figure. 

- - - - - - - - - - - - - - - - - -

3) A minor point, but there 10 superscript references in Table 1, presumably to some explanatory notes about the data point presented, but I could not find any explanation for them, either in the main text or the supplementary material. They should either be removed or (better) the explanatory notes should be provided.

Authors: These notes were inadvertently left out due to the complications of submitting large tables in a particular format. They are visible in the revised Table.

- - - - - - - - - - - - - - - - - -

4) I thought the discussion was a little timid. The work was done in early 2020, in the context of a pandemic that has dramatically underscored the need for good quality, consistent and easily available information from trial registries, partly to be able to track the numbers, types and results of trials relating to COVID-19, partly because public health decisions require a network of data sources at a global level and registries should be a key part of this. That point might have been worth including – improving trial registry systems has become more urgent!

Authors: We have rewritten the discussion, which include the following lines. 

“The ongoing Covid-19 pandemic has forcefully brought home the need for high quality trial registries with information that is consistent, comprehensive and available in a user-friendly fashion. Billions of people need to be immediately protected from the virus, and large numbers of drugs and vaccines are in trials. There is world-wide interest in these trials, and information that is being tracked includes what is being trialled; where are these trials taking place; and what are the results of these trials? Each country needs to take public health decisions, which will evolve as trials running in different parts of the world yield results. Public trial registries are one of the fastest ways of communicating these results.

Further, the publicly available, freely accessible information in trial registries helps to build trust with the public [26,44]. Covid-19 trials have been among the fastest recruiting trials in history [50,51], and it is possible that the publicly available information in trial registries has helped many of the potential trial participants decide to enrol.

It is not just that everyone is interested in the positive outcomes of trials. For example, an inspection of the CTRI records of hundreds of covid-19 trials being run in India has thrown up quality issues in almost all of them. Based on negative publicity, the government has taken action in some cases [52].”

- - - - - - - - - - - - - - - - - -

5) Similarly, although there is a general sentiment expressed that registries should improve, there were no concrete suggestions as to how this might be achieved or who needs to be involved, e.g. by greater collaboration between registries, perhaps orchestrated by the WHO, or by using the influence of funders and publishers to re-iterate the need for greater consistency. Are some of the aspects that were assessed easier to improve than others? If so how could they be progressed? Should there be a web page with a regularly updated 'score card' for the trial registries? I appreciate this was an initial survey but I think it might have been useful to venture, if only briefly, into this area in the discussion.

Authors: We have added the following lines to the Results and Discussion: 

There is a long history of various stakeholders arguing for the need to improve registries and the quality of trial registration. Examples include academics and health activists [53–55], journals (ICMJE) [56], WHO [41], registry managers [57], funders [58,59] and governments [60]. Each of these efforts has led to some improvements in the number and quality of trial records hosted by registries. However none of them has led to a perfect set of records. It is likely that the only way this will be achieved is if all stakeholders continue to apply pressure on the registries. Studies such as this one help to highlight deficiencies, which adds to the other efforts aimed at improving registries. Further, the authors would welcome other researchers’ efforts to create and update a website that lists the scorecard, with periodic updates. Should such a website not be created by any other group, the authors intend to re-evaluate the registries’ performance on the scorecard every few years.

- - - - - - - - - - - - - - - - - -

6) Another issue largely missing from the discussion: the authors mention that registries have many different types of users – researchers, clinicians, members of the public, data scientists, etc. I wonder if this should therefore lead to different scoring systems – perhaps with different weightings and / or items – for each of those major user groups. Those could provide additional insight into the strengths and weakness of different repositories, and thus more clearly identify areas of improvement, but could also be consolidated into an overall score if desired. For example, although the authors state their assumption is that most users would not have the technical expertise to use APIs, and / or scraping and crawling systems to retrieve data, the integration of trial registries with other data systems, and thus the ability to support bulk download by machines, is becoming increasingly important. I would have liked to have seen this aspect more explicitly included in any 'to do list' of possible future assessments, along with considerations of data quality, completeness, and the support for reporting results

Authors: In the Discussion, while enumerating the various kinds of users of registry data, we have stated that “Other categories of users, such as medical professionals, patients, trial sponsors, policy makers, data scientists and so on, may wish to alter the assessed features, or the scoring, in order to rank the registries according to their priorities. For instance, a data scientist would be very appreciative of ANZCTR, which specifically enables web crawling of its records [49]. Furthermore, the managers of other registries, either public or private, and either based on the data in the PR+ or not, may be interested in the results of this study.”. We do not feel confident of creating different scoring systems. Ideally, this should be done by polling at least a few individuals in each category of users, and we would find it extremely challenging to do this in India. As such, any additional scoring system that we developed would be based on unvalidated assumptions, and would be unconvincing, even to us.

- - - - - - - - - - - - - - - - - -

7) There is a minor but distracting typo in the first paragraph of the Results section (5,72,901)

Authors: We had used the Indian system. We have corrected this to 572,901.

- - - - - - - - - - - - - - - - - -

Having listed all of the points above I would re-iterate that overall I think the paper is useful and should be published. The points are offered as suggestions for possible improvement.

REVIEWER #2:

The concept is interesting but needs to be re-written. 

 1. The paper should first start with a good explanation of the origins of the ISCTR. For example, "following the Ministerial Summit on Health Research that took place in Mexico City, Mexico, in November 2004, participants called for the WHO to facilitate the establishment of: "a network of international clinical trials registers to ensure a single point of access and the unambiguous identification of trials".

Authors: We have rewritten the Introduction to include these events and further details of the ICTRP.

- - - - - - - - - - - - - - - - - -

 2. https://www.who.int/ictrp/about/en/ The authors need to be more complete in explaining the WHO registry network including primary vs partner registries as well as data providers; the differences of each. Then as it relates to registries what kind of papers have been published; findings; some of this is introduced at a high level in the discussion section which belongs in the introduction. 

Authors: We have include these points in the Introduction.

- - - - - - - - - - - - - - - - - -

 3. The authors are not clear in their terminology (for example, versions vs features). There is reference to WHO's 24-field Trial Registration Data Set vs 17 features vs 14 features selected by the authors; there is reference to the 9 standards; hence it is not clear how these "concepts" inter-relate (24 vs 17 vs 9; data set vs standards) and why the authors selected 14 features (which is perhaps more attributes than features).

Authors: We have now ensured the following:

(a) the word ‘section’ only refers to one or more of the nine sections of the ISCTR;

(b) the word ‘feature’ only refers to one or more of the 14 features of each registry that are the focus of this study; 

(c) since the different registries may have different variants of each ‘feature’, we use the word ‘variation’ in this context.

We have also included this list as a Box within Methods so that readers have no confusion regarding the terminology used.

- - - - - - - - - - - - - - - - - -

 4. How does WHO refer to author defined "features".

Authors: As described in Methods, the ‘features’ defined in this study have been compiled by the authors from different sources, including ISCTR. Therefore ISCTR discusses some, but not all, of these features directly. However, each feature maps to one or more standards set forth in ISCTR.

- - - - - - - - - - - - - - - - - -

 5. The score card is hard to follow; the scoring is not understandable (for example, "for features with multiple variants, the score ranges from 1 to 5" is not clear and then how scores were calculated: as in the case of chCTR for advanced search fields TRDS a score of 17 was assigned). 

Authors: We have reorganized the Methods section, and added further details. This includes portions from Box 1. We have also moved Table 2b to the methods section (where it is now Table 1) to make the rationale of the scorecard available upfront. Further, we have rewritten the Results and Discussion to include a more detailed analysis of the findings. We hope this alleviates the confusion around the scorecard.

- - - - - - - - - - - - - - - - - -

 6. The meaning of the scorecard is not clear; the interpretation of the findings are inconsistent and leaves the reader bewildered. 

Authors: We regret that the first version of the manuscript was so confusing. Please refer to our response to the comment before this (Comment 5). 

- - - - - - - - - - - - - - - - - -

 7. Please see specific comments in the attached word document. 

Authors: Please find below a response to each of the comments in the manuscript file, which we have numbered from 7.1 to 7.44). In each case we have referenced the line in the original pdf, where the reviewer’s comment has been taken from.

- - - - - - - - - - - - - - - - - -

7.1 Perhaps to elaborate on the 14 features briefly: what do they cover off on. (Line 31)

Authors: This has been done. [Lines 32–35, and 41–44 of the revised manuscript]

- - - - - - - - - - - - - - - - - -

7.2 Would not include limitations here. Only in the body of the paper under the proper section. (Line 42)

Authors: This has been done.

- - - - - - - - - - - - - - - - - -

7.3 New information should not be introduced in the conclusion; rather include in the results section. (Line 51)

Authors: This has been done. [Lines 587–589 of the revised manuscript]

- - - - - - - - - - - - - - - - - -

7.4 such as? Elaborate (Line 53)

Authors: This has been done. [Lines 63–68 of the revised manuscript]

- - - - - - - - - - - - - - - - - - 

7.5 elaborate on types of comments and analyses. (Line 67)

Authors: This has been done. [Lines 72–76 of the revised manuscript]

- - - - - - - - - - - - - - - - - -

7.6 awkwardly written. "Set up to facilitate"? The ICTRP was designed to help facilitate. (Line 68)

Authors: This has been reworded. [Lines 81–85 of the revised manuscript]

- - - - - - - - - - - - - - - - - -

7.7 replace wording with "not an ICTRP recognized registry"... (Line 71)

Authors: This has been reworded. [Lines 85–88 of the revised manuscript]

- - - - - - - - - - - - - - - - - - 

7.8 why were the other partner registries not included? or at least some of the other partner registries? (Line 74)

Authors: We have not included *any* partner registry in our analysis. This issue has been covered in more detailed now. [Lines 88–93 of the revised manuscript]

- - - - - - - - - - - - - - - - - - 

7.9 Write as Primary Registries Plus (PR+) (Line 74)

Authors: This has been done. [Lines 92–93 of the revised manuscript]

- - - - - - - - - - - - - - - - - -

7.10 Define the nine sections... (Line 77)

Authors: This has been done. [Lines 97–100 of the revised manuscript]

- - - - - - - - - - - - - - - - - - 

7.11 Recommend using 3rd person objective. Not "we" (Line 77)

Authors: It would be extremely challenging to rewrite the Methodology in 3rd person. However, we have ascertained that 'we' is used in the Methods' section, in articles that have appeared in well-known journals including PLOS ONE such as:

https://www.bmj.com/content/362/bmj.k3218

https://trialsjournal.biomedcentral.com/articles/10.1186/1745-6215-15-428
https://journals.plos.org/plosone/article?id=10.1371/journal.pone.0193088#sec006

We hope that it is alright if we leave the construction as it is.

- - - - - - - - - - - - - - - - - - 

7.12 not clear: using terms like versions, features, sections - difficult for the reader to follow (Line 83)

Authors: We have addressed this in our response to this Reviewer’s point 3.

- - - - - - - - - - - - - - - - - - 

7.13 important to note that not all fields in clinicaltrial.gov are mandatory. is this the case in other registries as well? (Line 96)

Authors: Yes, it is true that all fields are not mandatory in any registry. However we are only examining the *presence* of certain fields, not whether trialists have filled each of them.

- - - - - - - - - - - - - - - - - -

7.14 Authors? (Line 98)

Authors: We have rewritten this sentence and it now reads, ‘All analyses were performed by one author (NV) and verified by the other (GS).”

- - - - - - - - - - - - - - - - - - 

7.15 Is this part of Methods (Line 102)

Authors: We have formatted the manuscript to more clearly demarcate the different levels of headings.

- - - - - - - - - - - - - - - - -

7.16 needs to be defined "multiple variants, score ranges" (Line 106)

Authors: We have defined the variants and the score ranges in Table 1.

- - - - - - - - - - - - - - - - - - 

7.17 difficult to follow; confusing (Line 107)

Authors: We have moved Table 2b to the methods section (where it is now Table 1) to make the rationale of the scorecard available upfront. We have also illustrated our scoring system with examples early in the Methods section. Additionally, we wish to highlight that lines 128–130 describe how the authors selected the criteria for the scorecard, based on literature review but mainly focussing on the ISCTR guidelines.

- - - - - - - - - - - - - - - - - - 

7.18 perhaps to refer to as "attributes"? (Line 117)

Authors: We have determined that ‘feature’ and ‘attribute’ are synonyms. Since we have used ‘features’ throughout the manuscript, we preferred to stick to it. Also, as detailed in our response to the comment 7.12, above, we have rationalized our use of the word ‘features’ so that there is no confusion over its usage.

- - - - - - - - - - - - - - - - - - 

7.19 why 14 of the 17? (Line 118)

Authors: While revising the manuscript, we have removed the three features that were not assessed quantitatively. This leaves 14 features, all of which are in the scorecard. We believe that these changes have removed room for confusion on this point, and improved the readability of the paper.

- - - - - - - - - - - - - - - - - - 

7.20 we went from 14 of 17 to 24. Not sure how this ties back to the 9 standards introduced under Methods (Line 120)

Authors: As mentioned in point 7.19, above, we have removed the three features that were assessed quantitatively, and the final number is 14. 

Regarding the number 24: As described in the manuscript, one of the 14 features is the WHO’s Trial Registration Data Set, or TRDS, which in turn is composed of 24 distinct fields (as defined by ISCTR). 

- - - - - - - - - - - - - - - - - -

7.21 weight not weightage (Line 123)

Authors: This has been rewritten. [Lines 229–233 of the revised manuscript]

- - - - - - - - - - - - - - - - - - 

7.22 why is this a feature of the "registry" vs what is required by government in the local country. (Line 133)

Authors: In order to avoid possible confusion, we have rephrased the sentence to: “As a first step, it is important to know how many records the database holds. This number should be readily available, and we have therefore analysed the ease of accessing it.” 

- - - - - - - - - - - - - - - - - -

7.23 why were these two subjects highlighted especially as they were not noteworthy or substantive. (Line 136)

Authors: We have deleted these two features now. 

- - - - - - - - - - - - - - - - - - 

7.24 is this a server issue; connectivity; (Line 140)

Authors: We do not know the reason for the lag in loading RPCEC results. It seems to be at the RPCEC end, since other registries gave us no problem. In any case, we have deleted the entire feature now.

- - - - - - - - - - - - - - - - - - 

7.25 would think this is an important aspect to score; the logic to not score this as half of the registries did not contain is a shortcoming of the analysis as registries that do not have this attribute should receive a lower rating. Completion of variables is less about the registry vs the owner of the "data" (sponsor). (Line 140)

Authors: We have now included in the Results and Discussion our analysis of whether the registry provides the reason for trial termination. However, the information pertaining to this feature is already present in the Extra fields. Hence we have not scored this features separately, since that would result in it being double counted. 

- - - - - - - - - - - - - - - - - - 

7.26 why these attributes? Why is SSL important? (Line 154)

Authors: We have now included a more detailed description of why SSL is important and why we have included this feature. [Lines 471–481 of the revised manuscript]

- - - - - - - - - - - - - - - - - -

7.27 this is the sponsor's responsibility: QC (Line 166)

Authors: In the Results and Discussion section we have now described why even though QC is the sponsor’s responsibility it does not always do this, and how it is the duty of the registry to facilitate higher quality registrations. [Lines 496–501 of the revised manuscript]

- - - - - - - - - - - - - - - - - -

7.28 these points should have been provided in the introduction. (Line 181)

Authors: We have now included these points in the Introduction.

- - - - - - - - - - - - - - - - - -

7.29 there is no enough discussion on what has been done, why there is a gap and how this fills the gap in a meaningful way. (Line 189)

Authors: We have rewritten the Introduction and have discussed the work done so far, and how our study fills a lacuna in the analysis and reporting of registries’ performance. 

- - - - - - - - - - - - - - - - - - 

7.30 why is this ideal? (Line 195)

Authors: As described in the manuscript, we propose an ‘interim ideal’ registry based on the features that we have assessed. That it is a limited goal on the way to achieving everything that ISCTR requires. And it is the ‘ideal’ from amongst the various options that one or more of the registries are already using. 

- - - - - - - - - - - - - - - - - -

7.31 seems per line 180 to 189 there have been other studies. not clear. (Line 215)

Authors: We believe that the revised manuscript addresses this concern.

- - - - - - - - - - - - - - - - - - 

7.32 how was this impartial? (Line 216)

Authors: Primarily based on the recommendations of ISCTR, we determined which features of the registries to assess. All scoring rationales were also based on the minimum standards outlined in the ISCTR, and recommendations from earlier studies in our literature survey. We believe that our scoring is impartial since this protocol rules out scoring that may be biased in favour of, or against, any particular registry.

- - - - - - - - - - - - - - - - - - 

7.33 over reach as this is a subjective statement; only state represents the authors' perspective not that of others. (Line 220)

Authors: We have changed this sentence, which now reads “The scorecard above analyses features that are of interest to the authors and, by extension, possibly to other researchers concerned with the health of the trial ecosystem overall.” 

- - - - - - - - - - - - - - - - - - 

7.34 not clear (Line 222)

Authors: We have rephrased this to the following: “Other categories of users, such as medical professionals, patients, trial sponsors, policy makers, data scientists and so on, may wish to alter the assessed features, or the scoring, in order to rank the registries according to their priorities.” 

- - - - - - - - - - - - - - - - - - 

7.35 registries cannot be interested in something; only those who work on registries (Line 224)

Authors: We have rephrased the sentence to the following: “Further, the managers of other registries, either public or private, and either based on the data in the PR+ or not, may be interested in the results of this analysis.” 

- - - - - - - - - - - - - - - - - - 

7.36 less about the registry and more about how users utilize unless all fields are required by the registry. (Line 230)

Authors: We are not sure that we have understood this question. It is true that in no registry are all fields mandatory. Thus it would not be advisable to evaluate the quality of registration, and of information in a particular field, by comparisons across registries. However, as we have described in our response to Reviewer comment 7.13, we are only examining the presence of certain fields, not whether trialists have filled each of them. 

- - - - - - - - - - - - - - - - - - 

7.37 need to write using English proper language fit for scientific publication (Line 238)

Authors: We have rephrased the sentence to the following: “As noted above, 41% of the records are held in the other PR+, and they need to be examined as well.”

- - - - - - - - - - - - - - - - - - 

7.38 Define (Page 19, Box 1)

Authors: We have described our assessment of this feature in greater detail in the revised manuscript.

- - - - - - - - - - - - - - - - - - 

7.39 more importantly why do registries include something not recommended by ISCTR? (Page 20, Box 1)

Authors: We have added the following lines to the manuscript: “There may be a range of reasons for including these Extra fields. For instance, India had been criticized for the lack of appropriate oversight to ensure the ethical conduct of trials, and therefor CTRI asked trialists for details of the ethics committee even before ISCTR required this information [38,39]. Also, there have been demands from the Cochrane collaboration, and many other individuals and groups, to include several additional items in the ISCTR list, which WHO has not agreed to. It is alleged that the recommended list is closer to what industry demanded [40]. As such, although ISCTR may not list every field that many people believed to be essential, managers of particular registries may have chosen to list some of them.” 

- - - - - - - - - - - - - - - - - - 

7.40 Restructure: what does WHO require; what do registries do; where is the difference and impact on value of registries. (Page 20, Box1)

Authors: In each section of the Results and Discussion, we provide an introductory line, then the results, then the discussion. We have done this, incorporating the Reviewer’s points in this section. 

- - - - - - - - - - - - - - - - - - 

7.41 is this an author term? It is a prospective trial even if data is only entered after the study starts. (Page 21, Box1)

Authors: ‘Retrospective trials’ is used by other researchers – it is not our term. Nevertheless, earlier in the manuscript, we have added an explanatory line “Trials may be registered either prospectively or retrospectively, that is before the enrolment of the first participant or after.” Also, in order to avoid ambiguity, we have rephrased this sentence as follows: “Further, flagging retrospectively registered ones may shame the registrants into registering prospectively in future”.

- - - - - - - - - - - - - - - - - - 

7.42 what were the findings? (Page 21, Box1)

Authors: We have revised the text to include the findings of the analysis.

- - - - - - - - - - - - - - - - - - 

7.43 how was 3 assigned out of 5: what does a value of 1 or 2 or 3 or 4 or 5 represent? (Page 8, Table 2a)

Authors: As explained in the response to the Reviewer’s comment 5, we have moved Table 2b to the methods section (where it is now Table 1) to make the rationale of the scorecard available upfront. We hope this will alleviate the confusion regarding scoring. 

- - - - - - - - - - - - - - - - - - 

7.44 11? 17? thought scores were 1 to 5. (Page 8, Table 2a)

Authors: Please refer our response to the previous point (Reviewer’s comment 7.43)

- - - - - - - - - - - - - - - - - -

8. The topic is interesting but the paper needs to be rewritten and the concept of a scorecard has to be rethought to ensure it has logical relevance to the reader, that the scoring is understandable and can be interpreted thereby leading to actionable insights. 

Authors: We have rewritten and reorganized the manuscript, and we hope that these changes address the reviewer’s concerns. 

- - - - - - - - - - - - - - - - - -

9. The paper has to be placed in context of other relevant studies completed to date.

Authors: We have rewritten the Introduction to address this concern.

---

## [Decision Letter · Decision Letter 1]

11 Feb 2021

PONE-D-20-16268R1

A comparative analysis of important public clinical trial registries, and a proposal for an interim ideal one

PLOS ONE

Dear Dr. Saberwal,

Thank you for submitting your manuscript to PLOS ONE. After careful consideration, we feel that it has merit but does not fully meet PLOS ONE’s publication criteria as it currently stands. Therefore, we invite you to submit a revised version of the manuscript that addresses the points raised during the review process.

After the last review I recruited a 3rd reviewer for their opinion on the paper. Overall, the view is that this is an interesting and relevant paper. As one reviewer commented, this paper starts the conversation rather than ending it. One reviewer also pointed out that having a unified registry is vey important in this era of Covid-19. I think that it would be worth discussing the importance of this work with respect to the Covid-19 epidemic.

We look forward to receiving your revised manuscript.

Kind regards,

Dermot Cox

Academic Editor

PLOS ONE

Reviewers' comments:

Reviewer's Responses to Questions

**Comments to the Author**

1. If the authors have adequately addressed your comments raised in a previous round of review and you feel that this manuscript is now acceptable for publication, you may indicate that here to bypass the “Comments to the Author” section, enter your conflict of interest statement in the “Confidential to Editor” section, and submit your "Accept" recommendation.

Reviewer #1: All comments have been addressed

Reviewer #2: (No Response)

Reviewer #3: All comments have been addressed

2. Is the manuscript technically sound, and do the data support the conclusions?

Reviewer #1: Yes

Reviewer #2: No

Reviewer #3: Yes

3. Has the statistical analysis been performed appropriately and rigorously? 

Reviewer #1: N/A

Reviewer #2: N/A

Reviewer #3: Yes

4. Have the authors made all data underlying the findings in their manuscript fully available?

Reviewer #1: Yes

Reviewer #2: No

Reviewer #3: Yes

5. Is the manuscript presented in an intelligible fashion and written in standard English?

Reviewer #1: Yes

Reviewer #2: No

Reviewer #3: Yes

6. Review Comments to the Author

Reviewer #1: The paper is now far better organised and much easier to follow, and provides a very useful comparison of trial registries, at least in the dimensions described, and a good discussion of some of the issues involved.

One error in the edit (line 302) can be easily corrected.

Reviewer #2: Stopped reviewing content after a few pages; the content is too general, not descriptive enough; too many generalities are stated. The author needs to reconsider writing a laser-focused paper where the reader is taken step by step from purpose to approach to findings in a coherent, focused, understandable and 'supported by facts' approach.

Reviewer #3: A) Summary: This paper is a good starting point and should lead to a wider discussion on how clinical trial registries can be enhanced to offer more comprehensive and standardized features.

B) The document is well written, technically sound and the data provided in the supplementary files supports the conclusions. While the suggestions for an ideal clinical trial registry are very helpful, they are just a starting point for further debate within the clinical trial community. Interestingly in the era of Covid, the need for clinical trial registries that can be used for global clinical trials was not discussed.

C) While the discussion was good overall, the manuscript did lack detailed input from the different types of users as to the strengths and weaknesses of each of the registries reviewed. Such input would help identify the areas for improvement including how future registries need to support the wide range of individual users technical skills.

7. PLOS authors have the option to publish the peer review history of their article (what does this mean?). If published, this will include your full peer review and any attached files.

Reviewer #1: **Yes: **Steve Canham

Reviewer #2: No

Reviewer #3: No

---

## [Author Response · Author response to Decision Letter 1]

21 Mar 2021

Dear Dr. Cox,

Please find below a response to each of the comments in the manuscript file, which we have numbered from 1–6 (including 4.1–4.11). In each case we have referenced (a) the line number in the original pdf, which the reviewer’s comment refers to and (b) the line number in the revised manuscript.

With regards,

Gayatri Saberwal

- - - - - - - - - - - - - - - - - -

1. Editor:

One reviewer also pointed out that having a unified registry is vey important in this era of Covid-19. I think that it would be worth discussing the importance of this work with respect to the Covid-19 epidemic.

Authors: We have addressed this in point 5.

- - - - - - - - - - - - - - - - - -

2. Reviewer #1: 

One error in the edit (line 302) can be easily corrected.

Authors: Thank you for bringing this to our attention. We have rectified this error as follows [line 304 of the revised manuscript]:

ISCTR recommends that at the minimum, there must be a basic text search, as well asand it must be possible to searches within the interventions and conditions fields. 

Changed to

ISCTR recommends that at the minimum, there must be a basic text search, and it must be possible to conduct searches within the interventions and conditions fields. 

- - - - - - - - - - - - - - - - - -

3. Reviewer #2: Stopped reviewing content after a few pages; the content is too general, not descriptive enough; too many generalities are stated. The author needs to reconsider writing a laser-focused paper where the reader is taken step by step from purpose to approach to findings in a coherent, focused, understandable and 'supported by facts' approach.

Authors: We are sorry that the reviewer feels this way. We have done our best to answer his/her specific queries, listed below.

- - - - - - - - - - - - - - - - - -

4. Reviewer # 2’s comment bubbles in the manuscript.

4.1 why these 14 features? how were these 14 features from the total set? what is the total set?

 (Line 39)

Authors: As described in the manuscript, we have compiled the list of 14 features based on a literature survey to understand the limitations of the registries [lines 129–133 of the revised manuscript]. However, we have evaluated these 14 features within the framework of the International Standards for Clinical Trial Registries (ISCTR) set forth by the WHO. As such, the total set of features can be taken as the set of standards described in ISCTR.

- - - - - - - - - - - - - - - - - -

4.2 not appropriate wording for a scientific paper. (Line 43)

Authors: Thank you for bringing this to our notice. We have rectified this error as follows [line 40–43 of the revised manuscript]:

The assessed features include the nature of the content; the number and nature of fields to conduct a search; data download formats; the nature of the audit trail; the health condition category; the documentation available on a registry website; and so on. 

Changed to

The assessed features include the nature of the content; the number and nature of fields to conduct a search; data download formats; the nature of the audit trail; the health condition category; the documentation available on a registry website; etc.

- - - - - - - - - - - - - - - - - -

4.3 you are reporting scores without stating what was the scoring convention. (Line 44)

Authors: Thank you for this suggestion. We have now included, in the Abstract, a sentence about the scoring rationale, as follows:

“The registries received scores for their particular variation of a given feature based on a scoring rationale devised for each individual feature analysed.” [lines 43–45 of the revised manuscript]

- - - - - - - - - - - - - - - - - -

4.4 mere is not appropriate for scientific publications. (Line 50)

Authors: We have changed this [line 51–53 of the revised manuscript]. 

Even with this limited assessment, it is clear that some of the registries have much work to do, although a mere dozen improvements would significantly improve them.

Changed to 

Even with this limited assessment, it is clear that some of the registries have much work to do, although even a few improvements would significantly improve them.

- - - - - - - - - - - - - - - - - -

4.5 word choice calls? not appropriate for a scientific publication. (Line 54)

Authors: The word ‘call’ is used quite routinely in the sense that we have used it. For example: “The first quantitative data demonstrating publication bias in clinical trials—and clear call for trial registries—was published in 1986”. (Goldacre B (2015) How to Get All Trials Reported: Audit, Better Data, and Individual Accountability. PLoS Med 12(4): e1001821. doi:10.1371/journal.pmed.1001821.) Therefore, we have not changed the word.

- - - - - - - - - - - - - - - - - -

4.6 why a battle? what is the supporting documentation of a battle? (Line 58)

Authors: For over 2 decades there have been efforts to try and get trialists to register their trials, and report their results on time. It has been described as a battle, such as in the following lines: “And does the registration battle foreshadow how current data transparency efforts—calls for companies to go beyond releasing summaries of trial results to releasing patient-level data from trials—will unfold?” Miller, J. E. (2013). (How a Clinical Trial Registry Became a Symbol of Misinformation. Hastings Center Report, 43 (6), 11-12.) Therefore, we have not changed the word.

- - - - - - - - - - - - - - - - - -

4.7 conditions? do you mean disease areas or therapeutic areas? what is meant by an expanded access study? (Line 64)

Authors: We have modified this line and it now reads as [lines 63–66 of the revised manuscript] 

...researchers have utilized the data in at least a dozen other ways, such as (i) analyzing the conditions, the medical interventions, the sponsors and so on of Expanded Access Studies registered in the United States (US) [5]; 

changed to

...researchers have utilized the data in at least a dozen other ways, such as (i) analyzing characteristics such as the disease areas, medical interventions, and sponsors of Expanded Access Studies registered in the United States (US) [5];

We have not changed or explained ‘Expanded Access Studies’, since this is a standard category of clinical studies permitted in the United States. An example is “Open-label, expanded access study of taliglucerase alfa in patients with Gaucher disease requiring enzyme replacement therapy. Kuter DJ, et al. Blood Cells Mol Dis. 2020 May;82:102418. doi: 10.1016/j.bcmd.2020.102418”

- - - - - - - - - - - - - - - - - -

4.8 contravention of the law? (Line 67)

Authors: In this instance, we are referring to cases where trials are conducted in violation of the law, as explained in reference 9. 

- - - - - - - - - - - - - - - - - -

4.9 hidden duplicates? (Line 74)

Authors: The term hidden duplicates has been explained in references 15 and 16. It refers to a trial that is registered in two registries, but which cannot be unambiguously identified as duplicate entries of the same trial, because neither has listed the other’s ID in its ‘other study ID’ or ‘secondary ID’ field.

- - - - - - - - - - - - - - - - - -

4.10 Vague - why is the author making the statement? What advances? Writer needs to ground each statement in relevance of the argument being made with enough information for the reader to gain "usuable knowledge for the rest of the paper". (Line 75)

Authors: We have changed this [lines 76–79 of the revised manuscript]. 

Other studies have looked into the challenges faced, and advances made by individual registries [17,18].

changed to 

Other studies have looked into the evolution of individual registries – the challenges faced, and advances made [17,18]. Such analyses have improved our understanding of clinical trial registries globally, and reinforce the efforts for creating uniform standards for trial registration globally.

- - - - - - - - - - - - - - - - - -

4.11 what scandals? (line 77)

Authors: We have changed this [lines 81–83 of the revised manuscript]:

Certain high profile scandals [19,20] resulted in numerous calls to increase transparency in clinical trials and to improve the public’s trust in the trials enterprise. 

Changed to

Certain high profile scandals, such as the Vioxx case in 2004 and the Paxil case in 2015 [19,20], resulted in numerous calls to increase transparency in clinical trials and to improve the public’s trust in the trials enterprise. 

- - - - - - - - - - - - - - - - - -

5. Reviewer #3

Interestingly in the era of Covid, the need for clinical trial registries that can be used for global clinical trials was not discussed.

Authors: We would like to clarify that although each registry discussed in this work has a well-defined remit in terms of what kinds of trials it hosts, all of them – as far as we can determine – are open to registering global trials, provided their own country or region is part of such trials. Perhaps the reviewer is referring to a unified registry, either for all conditions, or specifically for Covid-19 trials? We have added the following as lines 571–588 of the revised manuscript.

“A handful of the registries evaluated in this study have provided a customized search for Covid-19 trials on their home pages. Covid-19 knowledge hubs, that facilitate access to trials that are registered with existing registries such as those described in this work, have also been set up [53–55]. Undoubtedly, such efforts to catalogue all Covid-19 trials, and experiences with Covid-19, has contributed to a more rapid understanding of the virus, and a more efficient management of the pandemic. 

This makes the case for (a) more of such knowledge hubs, or (b) a single world-wide trial registry. Arguments for a single world-wide registry have been made for many years [56]. However there are also counter-arguments, which highlight the fact that local registries (a) meet local ethics and regulatory requirements; (b) are easier to establish and monitor; (c) can be customized to local requirements; (d) are dependent on local infrastructure; (d) provide the national trial landscape and (e) can be a source of national pride [56,57].

Given that there are pros and cons to having a single world-wide registry, the best way forward may be to work to improve PR+ registries. If higher data quality and data sharing standards were more widespread, this would facilitate interoperability and data transfer while maintaining data integrity.

- - - - - - - - - - - - - - - - - -

6. While the discussion was good overall, the manuscript did lack detailed input from the different types of users as to the strengths and weaknesses of each of the registries reviewed. Such input would help identify the areas for improvement including how future registries need to support the wide range of individual users technical skills.

Authors: In the Discussion (line 544–550), while enumerating the various kinds of users of registry data, we have stated that “Registries have many users. The scorecard above analyses features that are of interest to the authors and, by extension, possibly to other researchers concerned with the health of the trial ecosystem overall. Other categories of users, such as medical professionals, patients, trial sponsors, policy makers, data scientists and so on, may wish to alter the assessed features, or the scoring, in order to rank the registries according to their priorities. For instance, a data scientist would be very appreciative of ANZCTR, which specifically enables web crawling of its records [49].”. We do not feel confident of creating different scoring systems. Ideally, this should be done by polling at least a few individuals in each category of users, and we would find it extremely challenging to do this in India. As such, any additional scoring system that we developed would be based on unvalidated assumptions, and would be unconvincing, even to us.

---

## [Editor Report · Decision Letter 2]

22 Apr 2021

A comparative analysis of important public clinical trial registries, and a proposal for an interim ideal one

PONE-D-20-16268R2

Dear Dr. Saberwal,

We’re pleased to inform you that your manuscript has been judged scientifically suitable for publication and will be formally accepted for publication once it meets all outstanding technical requirements.

Kind regards,

Dermot Cox

Academic Editor

PLOS ONE
---

## [Editor Report · Acceptance letter]

29 Apr 2021

PONE-D-20-16268R2 

A comparative analysis of important public clinical trial registries, and a proposal for an interim ideal one 

Dear Dr. Saberwal:

I'm pleased to inform you that your manuscript has been deemed suitable for publication in PLOS ONE. Congratulations! Your manuscript is now with our production department. 

Kind regards, 

on behalf of

Dr. Dermot Cox 

Academic Editor

PLOS ONE